# Metabolic Syndrome and Obesity-Related Indices Are Associated with Rapid Renal Function Decline in a Large Taiwanese Population Follow-Up Study

**DOI:** 10.3390/biomedicines10071744

**Published:** 2022-07-19

**Authors:** Wei-Yu Su, I-Hua Chen, Yuh-Ching Gau, Pei-Yu Wu, Jiun-Chi Huang, Yi-Chun Tsai, Szu-Chia Chen, Jer-Ming Chang, Shang-Jyh Hwang, Hung-Chun Chen

**Affiliations:** 1Department of General Medicine, Kaohsiung Medical University Hospital, Kaohsiung Medical University, Kaohsiung 807, Taiwan; s952135@gmail.com; 2Division of Endocrinology and Metabolism, Department of Internal Medicine, Kaohsiung Medical University Hospital, Kaohsiung Medical University, Kaohsiung 807, Taiwan; control521@gmail.com; 3Graduate Institute of Clinical Medicine, College of Medicine, Kaohsiung Medical University, Kaohsiung 807, Taiwan; cheesecaketwin@gmail.com; 4Division of Hematology and Oncology, Department of Internal Medicine, Kaohsiung Medical University Hospital, Kaohsiung 807, Taiwan; 5Department of Internal Medicine, Kaohsiung Municipal Siaogang Hospital, Kaohsiung Medical University, Kaohsiung 812, Taiwan; wpuw17@gmail.com (P.-Y.W.); karajan77@gmail.com (J.-C.H.); 6Division of Nephrology, Department of Internal Medicine, Kaohsiung Medical University Hospital, Kaohsiung Medical University, Kaohsiung 807, Taiwan; lidam65@yahoo.com.tw (Y.-C.T.); sjhwang@kmu.edu.tw (S.-J.H.); chenhc@kmu.edu.tw (H.-C.C.); 7Faculty of Medicine, College of Medicine, Kaohsiung Medical University, Kaohsiung 807, Taiwan; 8Research Center for Environmental Medicine, Kaohsiung Medical University, Kaohsiung 807, Taiwan

**Keywords:** metabolic syndrome, obesity-related indices, renal function decline, Taiwan Biobank, follow-up

## Abstract

A rapid decline in renal function can cause many complications, and therefore it is important to detect associated risk factors. Few studies have evaluated the associations among obesity-related indices and metabolic syndrome (MetS) with renal function decline. This longitudinal study aimed to explore these relationships in a large cohort of Taiwanese participants. The studied obesity-related indices were waist-to-height ratio (WHtR), A body shape index (ABSI), visceral adiposity index (VAI), lipid accumulation product (LAP), waist-to-hip ratio (WHR), body roundness index (BRI), conicity index (CI), body mass index (BMI), body adiposity index (BAI) and abdominal volume index (AVI). We included 122,068 participants in the baseline study, of whom 27,033 were followed for a median of four years. The baseline prevalence of MetS was 17.7%. Multivariable analysis showed that the participants with MetS and high VAI, WHtR, WHR, LAP, CI, BRI, BMI, BAI, AVI, and ABSI values were significantly associated with a high baseline estimated glomerular filtration rate (eGFR) (all *p* < 0.001). In addition, the participants with MetS (*p* < 0.001), high WHtR (*p* = 0.007), low LAP (*p* < 0.001), high BRI (*p* = 0.002), high CI (*p* = 0.002), high AVI (*p* = 0.001), high VAI (*p* = 0.017), and high ABSI (*p* = 0.013) were significantly associated with a low △eGFR, indicating a rapid decline in renal function. These results showed associations between MetS and high values of obesity-related indices except LAP with high baseline eGFR and rapid decline in kidney function. These findings suggest that screening for MetS and obesity may help to slow the decline in renal function in high-risk populations.

## 1. Introduction

The prevalence and incidence of end-stage renal disease (ESRD) continue to increase worldwide, including in Taiwan, which has the highest global prevalence of ESRD [1,2]. ESRD is associated with high rates of cardiovascular (CV) morbidity and mortality [3]. A rapid decline in renal function can cause many complications [4], and detecting the factors associated with a decline in renal function as early as possible is an important issue.

Metabolic syndrome (MetS) encompasses a range of metabolic abnormalities which are associated with increased risks of CV disease and diabetes mellitus (DM) [5]. The criteria for MetS were initially defined by the World Health Organization in 1998, and subsequent modifications to the criteria reflect increasing clinical evidence and analysis. The prevalence of MetS varies by geographical region, with an estimated global prevalence of around 25% and an age-standardized prevalence of 16.4% in Taiwan [6]. Anthropometric indices such as body mass index (BMI) and waist-to-height ratio (WHtR) are simple to calculate from easily obtainable factors such as waist circumference (WC), body weight (BW), hip circumference (HC), and body height (BH) [7].

Obesity is closely related to metabolic alteration in the body. According to a previous study, the mechanisms involved are related to the interaction between adipocytes and immune cells [8]. The increase in adipocyte size and the inability to store triglycerides would cause the development of an inflammatory response, which is thought to be associated with the formation of metabolic disorders. This would promote inflammation and activate apoptotic pathways. Inflammation of adipocytes affects insulin signaling and metabolism in adipocytes through the autocrine effects of inflammatory cells. Endocrine effects of adipokines affect insulin sensitivity in other tissues, resulting in local or even systemic insulin resistance [9]. A cohort study including 3,376,187 non-chronic kidney disease (CKD) subjects demonstrated that 8.1% of them had a rapid decline in kidney function, which was defined as a decrease in eGFR slope of >5 mL/min per 1.73 m^2^ [10]. The lowest risk of renal function decline was noted in patients with BMI between 25 to 30 kg/m^2^. Beyond this range, either an increase or decrease in BMI resulted in an increased rate of kidney function deterioration, showing a U-shaped association.

The exact mechanism underlying the association between MetS and obesity with kidney disease has yet to be fully elucidated, although insulin resistance has been proposed. Insulin resistance can cause renal injury through renin-angiotensin system activation, increases in aldosterone and angiotensin II with subsequent effects on endothelin-1 and insulin-like growth factor-1, and the generation of reactive oxygen species [11,12]. In addition, tissue resistance to insulin has been shown to lead to a reduction in nitric oxide production, which was associated with impaired tubuloglomerular feedback, hyperfiltration, and sodium retention, leading to disruption of the autoregulation of renal blood flow and glomerular filtration [13]. Obesity can also affect the production of adipokines, leading to renal dysfunction. Altered levels of adipokines, such as leptin, adiponectin, resistin, and visfatin, have been shown to decrease the GFR and increase albuminuria by increasing glomerular permeability, fusing podocytes, mesangial cell hypertrophy, and interfering with tubular networks [14]. Previous studies have reported associations between body roundness index (BRI) and WHtR with nonalcoholic fatty liver disease [15] and microalbuminuria and osteoporosis with MetS [16]. However, few studies have evaluated the relationships between MetS and obesity-related indices with a deterioration in renal function. Therefore, this longitudinal study aimed to explore these relationships in a large cohort of participants in the Taiwan Biobank (TWB).

## 2. Materials and Methods

### 2.1. The Taiwan Biobank

The Ministry of Health and Welfare, Taiwan, established the TWB, which was approved by its Ethics and Governance Council and the Institutional Review Board on Biomedical Science Research, Academia Sinica, Taiwan. It comprises data on lifestyle, genetic and medical factors for individuals without cancer, aged 30–70 years, and living in the community [17,18]. All participants enrolled in the TWB provide written informed consent, and then undergo interviews and physical examinations, and provide blood samples.

This study was approved by the Institutional Review Board of Kaohsiung Medical University Hospital (KMUHIRB-E(I)-20210058) and followed the Declaration of Helsinki. We included 122,068 participants in the baseline study and followed 27,033 of them for a median of four years (Figure 1). Associations between baseline MetS and obesity-related indices with baseline eGFR were analyzed in the baseline study (*n* = 122,068). In addition, associations between baseline MetS and obesity-related indices with eGFR decline (△eGFR) were analyzed in the follow-up study (*n* = 27,033).

### 2.2. Collection of Study Variables

During the physical examinations, systolic blood pressure (SBP), diastolic blood pressure (DBP), HC, BH, WC, and BW were recorded. During the interviews, the participants completed questionnaires, and data on lifestyle factors (such as smoking history, and exercise), personal medical history (such as hypertension and DM), age, and sex were obtained. Regular exercise was defined as doing any of the following for >30 min ≥3 times per week: swimming, any sport, exercise-based apps, jogging, hiking, cycling, and yoga. Work-related activity was not included [19].

Data on high-density lipoprotein cholesterol (HDL-C), low-density lipoprotein cholesterol (LDL-C), total cholesterol, triglycerides (TGs), hemoglobin, fasting glucose, uric acid, and estimated glomerular filtration rate (eGFR; which was calculated as reported previously [20]) were also collected.

### 2.3. Definitions of Renal Function Decline (△eGFR) and MetS

Renal function decline (△eGFR) was calculated as the eGFR at the end of follow-up minus that at baseline.

In this study, MetS were defined according to the modified NCEP-ATP III criteria for Asians as participants with ≥3 of the following [21]: (1) hyperglycemia, defined as fasting blood glucose concentration ≥ 100 mg/dL or a diagnosis of DM; (2) HDL-C < 50 mg/dL in women and <40 mg/dL in men; (3) TG concentration ≥ 150 mg/dL; (4) SBP ≥ 130 mmHg or DBP ≥ 85 mmHg, a diagnosis of hypertension, or prescriptions for anti-hypertensive medications; (5) abdominal obesity, defined as a WC in ≥80 cm in women and ≥90 cm in men.

### 2.4. Calculation of Obesity-Related Indices

The calculation of obesity-related indices is shown in Table 1.

### 2.5. Statistical Analysis

SPSS version 19.0 was used for all analyses (SPSS Inc., Chicago, IL, USA). Categorical variables were presented as frequencies (percentage) and compared using the chi-squared test. Continuous variables were presented as mean (±standard deviation) and compared using the independent *t*-test. Multivariable linear regression analysis with significant variables from the univariable analysis was used to identify associations among obesity-related indices and MetS with baseline eGFR and △eGFR. *p*-values < 0.05 was considered to be statistically significant.

### 2.6. Comparisons of Clinical Characteristics between the Participants with and without MetS at Baseline

A total of 122,068 participants were enrolled (43,849 males, 78,219 females, mean age 49.9 ± 10.9 years) and divided into two groups according to whether they had (*n* = 100,519; 82.3%) or did not have MetS at baseline (*n* = 21,549; 17.7%).

Compared to the non-MetS group, the MetS group were older, predominantly male, had higher prevalence rates of DM and hypertension, and higher SBP, DBP, BH, BW, WC, HC, fasting glucose, hemoglobin, TGs, total cholesterol, LDL-C, and uric acid, and lower eGFR and HDL-C (Table 2). Regarding obesity-related indices, the participants with MetS had higher values of all the studied obesity-related indices (VAI, WHtR, WHR, LAP, CI, BRI, BMI, BAI, AVI, and ABSI).

## 3. Results

### 3.1. Association between MetS and Obesity-Related Indices with Baseline eGFR in All Participants

The results of univariable linear regression analysis of the factors associated with baseline eGFR in the whole cohort (*n* = 122,068) showed that male sex, older age, hypertension, DM, smoking history, and high values of uric acid, fasting glucose, SBP, DBP, hemoglobin, TGs, total cholesterol, and LDL-C, and were associated with a low baseline eGFR (Table 3). In addition, low HDL-C was also associated with a low baseline eGFR.

The results of the multivariable linear regression analysis to determine the associations among MetS and obesity-related indices with baseline eGFR in the whole cohort are shown in Table 4. The following models were used:(1)Adjusted for sex, age, smoking history, hemoglobin, LDL-C, total cholesterol, and uric acid (the significant factors in univariable analysis excluding DM, hypertension, SBP, DBP, fasting glucose, TGs, and HDL-C) for MetS.(2)Adjusted for sex, age, smoking history, DM, hypertension, SBP, DBP, uric acid, fasting glucose, hemoglobin, TGs, HDL-C, LDL-C, and total cholesterol (significant factors in univariable analysis) for WHtR, WHR, CI, BRI, BMI, BAI, AVI, and ABSI.(3)Adjusted for sex, age, smoking history, DM, hypertension, SBP, DBP, uric acid, fasting glucose, hemoglobin, HDL-C, LDL-C, and total cholesterol (significant factors in the univariable analysis except for TGs) for LAP.(4)Adjusted for sex, age, smoking history, DM, hypertension, SBP, DBP, uric acid, fasting glucose, hemoglobin, LDL-C, and total cholesterol (significant factors in the univariable analysis except for TGs and HDL-C) for VAI.

After multivariable analysis, participants with MetS and high VAI, WHtR, WHR, LAP, CI, BRI, BMI, BAI, AVI, and ABSI were significantly associated with a high baseline eGFR (all *p* < 0.001).

### 3.2. Association between MetS and Obesity-Related Indices with △eGFR in Follow-Up Participants

The results of univariable linear regression analysis of the factors associated with △eGFR in follow-up participants (*n* = 27,033) showed that female sex, older age, hypertension, DM, high values of SBP, DBP, and fasting glucose, and low values of uric acid, hemoglobin, total cholesterol, HDL-C, and LDL-C were associated with low △eGFR (Table 5).

The results of the multivariable linear regression analysis to determine the associations among MetS and obesity-related indices with △eGFR in the follow-up participants are shown in Table 6. The following models were used:(1)Adjusted sex, age, hemoglobin, LDL-C, total cholesterol, and uric acid (the significant factors in univariable analysis excluding DM, hypertension, SBP, DBP, fasting glucose, and HDL-C) for MetS.(2)Adjusted for sex, age, DM, hypertension, SBP, DBP, uric acid, fasting glucose, hemoglobin, HDL-C, LDL-C, and total cholesterol (significant factors in univariable analysis) for WHtR, WHR, CI, BRI, BMI, BAI, AVI, and ABSI.(3)Adjusted for sex, age, DM, hypertension, SBP, DBP, uric acid, fasting glucose, hemoglobin, LDL-C, and total cholesterol (significant factors in the univariable analysis except for HDL-C) for VAI.

The results of the multivariable analysis showed that MetS (*p* < 0.001), high WHtR (*p* = 0.007), low LAP (*p* < 0.001), high BRI (*p* = 0.002), high CI (*p* = 0.002), high AVI (*p* = 0.001), high VAI (*p* = 0.017), and high ABSI (*p* = 0.013) were significantly associated with low △eGFR. However, BMI (*p* = 0.075), WHR (*p* = 0.126), and BAI (*p* = 0.066) were not associated with △eGFR.

## 4. Discussion

In this study, we analyzed associations among MetS, obesity-related indices, baseline eGFR, and △eGFR in 122,068 participants at baseline and 27,033 after four years of follow-up. After adjusting for confounders, the analysis showed that the participants with MetS and high values of obesity-related indices were associated with high baseline eGFR and low △eGFR, indicating a rapid decline in renal function.

There are several important findings in this study. First, the participants with MetS were associated with a high baseline eGFR after multivariable analysis. In contrast, previous studies on patients with and without CKD have reported that those with MetS had a lower eGFR at baseline than those without MetS [29,30,31,32]. A possible explanation for this discrepancy may be due to glomerular hyperfiltration. An elevation in GFR has been observed in some patients in the early stages of DM. This elevation has been attributed to an increase in glomerular hydraulic pressure and transcapillary convective flux of macromolecules and ultrafiltrate, augmented by increases in extracellular volume and sodium reabsorption in the proximal tubules and altered tubuloglomerular feedback [33,34]. Tomaszewski et al. demonstrated an association between an increase in absolute estimated creatinine clearance and the cumulative number of MetS components, and a 6.9-fold higher risk of glomerular hyperfiltration in participants with MetS than in those without [35]. Several adipocytokines have been associated with changes in GFR [14], suggesting that adipocytokines may mediate the association between increased adiposity in MetS and glomerular hyperfiltration, which may then lead to glomerular dysfunction.

Second, MetS was associated with low △eGFR, indicating a rapid decline in kidney function. A large 10-year prospective cohort study demonstrated a greater risk of a rapid decline in eGFR in participants with MetS than in those without MetS [36]. In addition, a Chinese community-based study reported that the presence of MetS and more MetS components were significantly correlated with a rapid deterioration in eGFR [29]. In contrast, Cheng et al. reported that MetS was not associated with a rapid decline in kidney function in older adults but that a one-unit increase in insulin resistance was associated with a 1.16-fold increase in the hazard ratio of a decline in renal function in older adults without DM [37]. The exact mechanism underlying the association between MetS and kidney disease has yet to be fully elucidated, although insulin resistance has been proposed. Insulin resistance can cause renal injury through renin-angiotensin system activation, increases in aldosterone and angiotensin II with subsequent effects on endothelin-1 and insulin-like growth factor-1, and the generation of reactive oxygen species [11,12]. In addition, tissue resistance to insulin has been shown to lead to a reduction in insulin-dependent responses of the phosphoinositide 3-kinase/protein kinase B pathway, leading to a reduction in nitric oxide production and subsequently impaired nitric oxide-dependent vascular relaxation, increased tissue inflammation, and fibrosis.

Moreover, a reduction in nitric oxide production has been associated with impaired tubuloglomerular feedback, hyperfiltration, and sodium retention, leading to disruption of the autoregulation of renal blood flow and glomerular filtration [13]. In addition to the mechanisms involving systemic inflammation and oxidative stress, “cardiometabolic memory” has been proposed to explain how transient intensive lowering of glucose can induce “memory” to suppress diabetic microangiopathies, resulting in persistent benefits even after ceasing treatment [38]. Possible mechanisms include the non-enzymatic glycation of cellular proteins and lipids along with the excess production of cellular reactive nitrogen and oxygen species, which together maintain stress signaling.

Third, all 10 studied obesity-related indices were positively correlated with baseline eGFR. A prospective population-based longitudinal analysis conducted in Japan reported an association between high VAI and lower baseline eGFR [39]. Another study reported an association between high LAP and VAI with lower baseline eGFR in adults without DM [40], and Kim et al. found a negative association between ABSI and eGFR in a cohort of older adults [41]. In the study by Ou et al., high AVI, BMI, BRI, CI, LAP, WHR, WHtR, and VAI were significantly associated with advanced CKD (eGFR < 30 mL/min/1.73 m^2^) in patients with type 2 DM, whereas BAI and ABSI were not [42]. Furthermore, Chung et al. reported a higher baseline eGFR in diabetic patients with high BMI even after adjusting for age [43]. Aging and obesity lead to muscle atrophy, and a recent animal study demonstrated that the accumulation of perimuscular adipose tissue accelerated muscle fiber atrophy by increasing proteolysis and cellular senescence [44]. Muscle mass has been independently correlated with serum creatinine [45], which is a significant determinant when estimating GFR using the four-variable Modification of Diet in Renal Disease study equation [18]. Therefore, people with obesity and decreased muscle mass may have higher baseline eGFR due to a low serum creatinine level.

Fourth, high values of the studied obesity-related indices, except BMI, WHR, BAI, and LAP, were associated with low △eGFR. Visceral adiposity has been shown to play a significant role in the pathogenic pathways involved in MetS [46], and obesity-related indices have been associated with MetS [47]. Previous studies have reported an association between high VAI with a higher risk of progressing to CKD [39,40], and high WC and WHtR have been associated with a faster decline in GFR [48]. Nevertheless, a recent study showed that LAP was not significantly associated with renal function decline in adults without DM [40]. A meta-analysis of 39 general population cohorts with a total of 5,459,014 adults from 1970 to 2017 found that the respective hazard ratios for a decline in GFR comparing BMI 35, and 40 with BMI 25 were 1.28 (95% confidence interval 1.14 to 1.44) and 1.46 (1.28 to 1.67), respectively, after adjusting for age, sex, race, current smoking, and other comorbidities [48]. However, in the same study, among patients with CKD, the risk of a decline in GFR was higher in those with a BMI of 35 or 20 compared to those with a BMI of 25. This nonlinear correlation may explain why BMI was not significantly associated with low △eGFR in our study. BMI is limited as it cannot distinguish between body muscle and fat mass or between visceral and subcutaneous adipose tissue, resulting in an underestimation of the metabolic risk associated with adiposity [40]. A previous longitudinal study of adults with baseline eGFR ≥ 60 mL/min/1.73 m^2^ without diabetes showed that WHR was not associated with a rapid decline in eGFR [49], which is consistent with our results.

Another interesting finding of this study is that high LAP was associated with high-baseline eGFR. However, unlike the other obesity-related indices, it was also positively correlated with △eGFR, indicating that high LAP had a protective effect on renal function. Yan et al. reported positive associations between LAP quartiles and the risk of CKD in community-dwelling Chinese female adults [50], and another cross-sectional study also showed similar findings, with a 2.32-fold increased risk of renal dysfunction in those with the highest LAP quartile compared to the lowest quartile [51]. In addition, a study of 6693 non-diabetic adults who were followed for 8.6 years reported that a higher rate of decline in renal function was associated with higher LAP before adjusting for confounders. However, the association was lost after adjusting for age, smoking, physical activity, systolic BP, fasting plasma glucose, and 2 h post-challenge plasma glucose [40]. The positive correlation between high LAP and high baseline eGFR may be explained by glomerular hyperfiltration, as mentioned above.

However, the mechanisms underlying the association between LAP and △eGFR are not completely understood. Compared to VAI, which also includes the components of WC and TG, LAP is a relatively simple index that is not corrected by BMI and HDL. Interestingly, we found paradoxical results from the aspect of insulin resistance [13]. A previous study reported that LAP had a greater impact on insulin resistance than BMI and WC in non-diabetic participants [51]; however, further studies with a longer follow-up are needed to clarify the relationship between LAP and △eGFR.

The strengths of the present investigation include the large population-based study cohort and follow-up of the associations among MetS, 10 obesity-related indices, and renal function. However, several limitations should also be noted. First, certain medications such as anti-diabetic, anti-hypertensive, and lipid-lowering agents can affect the development or prevention of MetS, lipid profile, fasting glucose and BP; however, we did not have data on these medications. This may have caused the associations between renal function and MetS to be underestimated. Second, the data on factors that could cause a rapid decline in renal function, such as proteinuria, were also unavailable. Third, participants enrolled in the TWB are of Han Chinese ethnicity, which may limit the application of our findings to other ethnicities. Finally, there may have been sample bias, as approximately 25% of TWB enrollees returned for follow-up assessments.

Reducing the decline in renal function to stop the progression of CKD is a clinically important issue. Because our study showed that MetS and obesity-related indices are associated with the rapid deterioration of renal function, we hypothesized that identifying and treating MetS and obesity may help to slow down the deterioration of renal function. A small, randomized control trial enrolled participants with MetS and no pre-existing renal dysfunction and assigned them to three groups: dietary weight loss, weight loss combined with aerobic exercise, and no treatment [52]. The results showed that weight loss helped to improve renal function, and the effect of combined aerobic exercise was better than dietary weight loss alone. Another retrospective study involved 13,310,924 subjects without ESRD who underwent two health examinations over two years [53]. The hazard ratio and 95% confidence interval of the cohort that consistently had MetS progressing to ESRD was 5.65 (95% confidence interval, 5.42–5.89) compared with the cohort that consistently did not have MetS.

In contrast, the hazard ratio of those with MetS who were followed up and found to have no MetS was 2.28 (2.15–2.42) compared with those with no MetS at all times. These studies may suggest that identifying and treating MetS and obesity may help improve the prognosis of the kidney. Combining our findings with WHtR, LAP, BRI, CI, AVI, VAI, and ABSI to identify patients with potentially rapidly deteriorating renal function in the obese population will allow clinicians to develop prevention strategies earlier.

In conclusion, we found that MetS and high values of obesity-related indices were associated with a high baseline eGFR and rapid decline in kidney function in our large cohort of Taiwanese adults. Our findings suggest that screening for MetS and obesity may help to slow the decline in renal function in high-risk populations. Further studies about new and useful predictive and prognostic biomarkers are needed to verify these findings in the future.

## Figures and Tables

**Figure 1 biomedicines-10-01744-f001:**
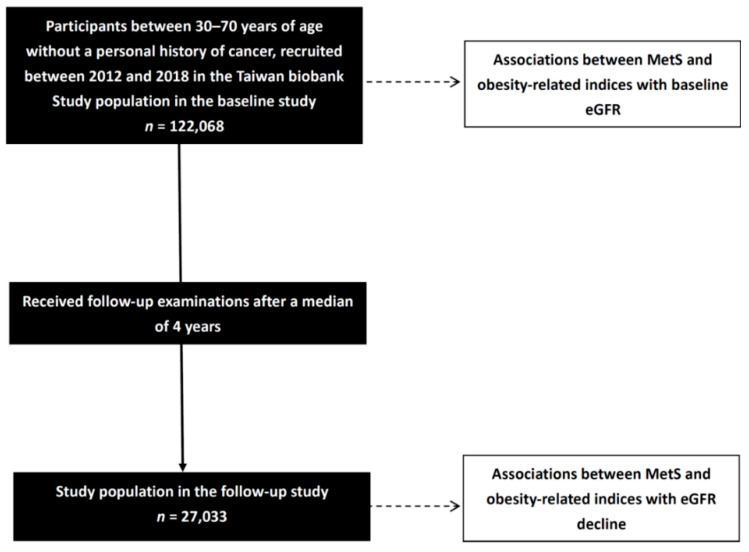
Flowchart of the study population.

**Table 1 biomedicines-10-01744-t001:** Calculation formula of each obesity-related index.

Item	Calculation Formula
BMI	BW (kg)/BH^2^ (m)
WHR	WC (cm)/HC (cm)
WHtR	WC (cm)/BH (cm)
LAP	$\left({\mathrm{WC}}_{\left(\mathrm{cm}\right)}-65\right)\times {\;\mathrm{TG}}_{\left(\mathrm{mmol}/L\right)}$$\;\mathrm{in}\;\mathrm{males},\;\mathrm{and}\;\left({\mathrm{WC}}_{\left(\mathrm{cm}\right)}-58\right)\times {\;\mathrm{TG}}_{\left(\mathrm{mmol}/L\right)}$ in females [22]
BRI	$364.2-365.5\times \sqrt{1-{\left(\frac{\frac{{\mathrm{WC}}_{\left(m\right)}}{2\pi }}{0.5\times {\mathrm{BH}}_{\left(m\right)}}\right)}^{2}\;\;}$ [23]
CI	$\frac{{\mathrm{WC}}_{\left(m\right)}}{0.109\times \sqrt{\frac{{\mathrm{BW}}_{\left(\mathrm{kg}\right)}}{{\mathrm{BH}}_{\left(m\right)}}}\;}\;$ [24]
BAI	$\frac{{\mathrm{HC}}_{\left(\mathrm{cm}\right)}}{{\mathrm{BH}}_{\left(m\right)}{}^{3/2}}-18\;$ [25]
AVI	$\frac{2\times {\left({\;\mathrm{WC}}_{\left(\mathrm{cm}\right)}\;\right)}^{2}+0.7\times {\left({\;\mathrm{WC}}_{\left(\mathrm{cm}\right)}-{\mathrm{HC}\;}_{\left(\mathrm{cm}\right)}\right)}^{2}}{1000}\;$ [26]
VAI	$\left(\frac{{\mathrm{WC}}_{\left(\mathrm{cm}\right)}}{39.68+\left(1.88\times \;\mathrm{BMI}\right)}\right)\times \left(\frac{{\mathrm{TG}}_{\left(\mathrm{mmol}/L\right)}}{1.03}\right)\times \left(\frac{1.31}{\mathrm{HDL}-C_{\left(\mathrm{mmol}/L\right)}}\right)$$\mathrm{in}\;\mathrm{males},\;\mathrm{and}\;\left(\frac{{\mathrm{WC}}_{\left(\mathrm{cm}\right)}}{36.58+\left(1.89\times \;\mathrm{BMI}\right)}\right)\times \left(\frac{{\mathrm{TG}}_{\left(\mathrm{mmol}/L\right)}}{0.81}\right)\times \left(\frac{1.52}{\mathrm{HDL}-C_{\left(\mathrm{mmol}/L\right)}}\right)$ in females [27]
ABSI	WC (m)/[BMI^2/3^(kg/m^2^) × BH^1/2^(m)] [28]

Abbreviations: BMI—body mass index; WHR—waist-to-hip ratio; WHtR—waist-to-height ratio; LAP—lipid accumulation product; BRI—body roundness index; CI—conicity index; BAI—body adiposity index; AVI—abdominal volume index; VAI—visceral adiposity index; ABSI—A body shape index.

**Table 2 biomedicines-10-01744-t002:** Comparison of clinical characteristics among participants according to baseline MetS.

Characteristics	MetS (−) (*n* = 100,519)	MetS (+) (*n* = 21,549)	*p*
Age (year)	49.1 ± 11.0	53.6 ± 10.2	<0.001
Male gender (%)	34.5	42.8	<0.001
DM (%)	2.2	19.1	<0.001
Hypertension (%)	7.5	31.6	<0.001
Smoking history (%)	25.8	33.9	<0.001
SBP (mmHg)	116.8 ± 16.9	132.6 ± 18.2	<0.001
DBP (mmHg)	71.8 ± 10.4	80.5 ± 11.4	<0.001
Body height (cm)	161.8 ± 8.2	162.3 ± 8.8	<0.001
Body weight (kg)	61.8 ± 11.6	73.0 ± 13.9	<0.001
Waist circumference (cm)	81.3 ± 9.3	92.7 ± 9.2	<0.001
Hip circumference (cm)	95.0 ± 6.6	100.5 ± 7.8	<0.001
Laboratory parameters			
Fasting glucose (mg/dL)	92.7 ± 13.7	110.9 ± 35.9	<0.001
Hemoglobin (g/dL)	13.7 ± 1.6	14.2 ± 1.6	<0.001
Triglyceride (mg/dL)	95.9 ± 59.1	207.6 ± 153.1	<0.001
Total cholesterol (mg/dL)	195.1 ± 35.0	198.3 ± 39.5	<0.001
HDL cholesterol (mg/dL)	57.1 ± 13.0	42.8 ± 8.6	<0.001
LDL cholesterol (mg/dL)	120.6 ± 31.2	122.3 ± 34.1	<0.001
eGFR (mL/min/1.73 m^2^)	110.9 ± 25.1	104.5 ± 26.9	<0.001
Uric acid (mg/dL)	5.3 ± 1.4	6.2 ± 1.5	<0.001
Obesity-related indices			
BMI (kg/m^2^)	23.5 ± 3.4	27.6 ± 3.9	<0.001
WHR (%)	85.4 ± 6.5	92.2 ± 6.0	<0.001
WHtR (%)	50.3 ± 5.6	57.2 ± 5.5	<0.001
LAP	23.5 ± 18.7	73.3 ± 56.3	<0.001
BRI	6.4 ± 1.7	8.6 ± 1.9	<0.001
CI	1.21 ± 0.08	1.27 ± 0.07	<0.001
BAI	28.3 ± 4.0	30.8 ± 4.6	<0.001
AVI	13.5 ± 3.1	17.4 ± 3.6	<0.001
VAI	1.3 ± 1.1	3.7 ± 3.4	<0.001
ABSI	0.078 ± 0.004	0.080 ± 0.005	<0.001

Abbreviations. DM—diabetes mellitus; SBP—systolic blood pressure; DBP— diastolic blood pressure; HDL—high-density lipoprotein; LDL—low-density lipoprotein; eGFR—estimated glomerular filtration rate; BMI—body mass index; WHR—waist-to-hip ratio; WHtR—waist-to-height ratio; LAP—lipid accumulation product; BRI—body roundness index; CI—conicity index; BAI—body adiposity index; AVI—abdominal volume index; VAI—visceral adiposity index; ABSI—A body shape index.

**Table 3 biomedicines-10-01744-t003:** Determinants for baseline eGFR using univariable linear regression analysis in all study participants (*n* = 122,068).

Characteristics	Univariable	
	Unstandardized Coefficient β (95% Confidence Interval)	*p*
Age (per 1 year)	−0.682 (−0.695, −0.670)	<0.001
Female (vs. male)	15.777 (15.491, 16.062)	<0.001
DM	−5.852 (−6.498, −5.206)	<0.001
Hypertension	−12.369 (−12.800, −11.937)	<0.001
Smoking history	−8.112 (−8.431, −7.794)	<0.001
SBP (per 1 mmHg)	−0.298 (−0.306, −0.290)	<0.001
DBP (per 1 mmHg)	−0.443 (−0.456, −0.431)	<0.001
Laboratory parameters		
Fasting glucose (per 1 mg/dL)	−0.072 (−0.079, −0.065)	<0.001
Hemoglobin (per 1 g/dL)	−3.864 (−3.951, −3.777)	<0.001
Triglyceride (per 1 mg/dL)	−0.031 (−0.033, −0.029)	<0.001
Total cholesterol (per 1 mg/dL)	−0.071 (−0.075, −0.067)	<0.001
HDL cholesterol (per 1 mg/dL)	0.216 (0.205, 0.227)	<0.001
LDL cholesterol (per 1 mg/dL)	−0.082 (−0.086, −0.077)	<0.001
Uric acid (per 1 mg/dL)	−7.101 (−7.193, −7.009)	<0.001

Values expressed as unstandardized coefficient β and 95% confidence interval. Abbreviations are the same as in Table 1.

**Table 4 biomedicines-10-01744-t004:** Association of MetS and obesity-related indices with baseline eGFR using multivariable linear regression analysis in all study participants (*n* = 122,068).

Obesity-Related Indices	Multivariable	
	Unstandardized Coefficient β (95% Confidence Interval)	*p*
MetS ^a^	2.190 (1.848, 2.532)	<0.001
BMI (per 1 kg/m^2^) ^b^	0.245 (0.205, 0.284)	<0.001
WHR (per 1%) ^b^	0.384 (0.362, 0.407)	<0.001
WHtR (per 1%) ^b^	0.427 (0.403, 0.451)	<0.001
LAP (per 1) ^c^	0.042 (0.036, 0.048)	<0.001
BRI (per 1) ^b^	1.149 (1.074, 1.225)	<0.001
CI (per 0.1) ^b^	3.197 (3.029, 3.365)	<0.001
BAI (per 1) ^b^	0.473 (0.437, 0.509)	<0.001
AVI (per 1) ^b^	0.425 (0.381, 0.468)	<0.001
VAI (per 1) ^d^	0.567 (0.495, 0.639)	<0.001
ABSI (per 0.01) ^b^	4.722 (4.452, 4.992)	<0.001

Values expressed as unstandardized coefficient β and 95% confidence interval. Abbreviations are the same as in Table 1. ^a^ Adjusted for age, sex, smoking history, hemoglobin, total cholesterol, LDL cholesterol, and uric acid (significant variables of Table 2 except for diabetes, hypertension, systolic and diastolic blood pressures, fasting glucose, triglyceride, and HDL cholesterol). ^b^ Adjusted for age, sex, diabetes, hypertension, smoking history, systolic and diastolic blood pressures, fasting glucose, hemoglobin, triglyceride, total cholesterol, HDL cholesterol, LDL cholesterol, and uric acid (significant variables of Table 2). ^c^ Adjusted for age, sex, diabetes, hypertension, smoking history, systolic and diastolic blood pressures, fasting glucose, hemoglobin, total cholesterol, HDL cholesterol, LDL cholesterol, and uric acid (significant variables of Table 2 except for triglyceride). ^d^ Adjusted for age, sex, diabetes, hypertension, smoking history, systolic and diastolic blood pressures, fasting glucose, hemoglobin, total cholesterol, LDL cholesterol, and uric acid (significant variables of Table 2 except for triglyceride and HDL cholesterol).

**Table 5 biomedicines-10-01744-t005:** Determinants for △eGFR using univariable linear regression analysis in follow-up participants (*n* = 27,033).

Characteristics	Univariable	
	Unstandardized Coefficient β (95% Confidence Interval)	*p*
Age (per 1 year)	−0.058 (−0.079, −0.038)	<0.001
Female (vs. male)	−0.699 (−1.148, −0.249)	0.002
DM	−4.052 (−5.011, −3.093)	<0.001
Hypertension	−1.842 (−2.479, −1.204)	<0.001
Smoking history	0.341 (−0.152, 0.833)	0.175
SBP (per 1 mmHg)	−0.067 (−0.079, −0.055)	<0.001
DBP (per 1 mmHg)	−0.051 (−0.071, −0.032)	<0.001
Laboratory parameters		
Fasting glucose (per 1 mg/dL)	−0.034 (−0.045, −0.023)	<0.001
Hemoglobin (per 1 g/dL)	0.444 (0.305, 0.582)	<0.001
Triglyceride (per 1 mg/dL)	0 (−0.003, 0.002)	0.751
Total cholesterol (per 1 mg/dL)	0.009 (0.003, 0.015)	0.004
HDL cholesterol (per 1 mg/dL)	0.019 (0.003, 0.036)	0.020
LDL cholesterol (per 1 mg/dL)	0.022 (0.015, 0.029)	<0.001
Uric acid (per 1 mg/dL)	1.061 (0.910, 1.211)	<0.001

Values expressed as unstandardized coefficient β and 95% confidence interval. Abbreviations are the same as in Table 1.

**Table 6 biomedicines-10-01744-t006:** Association of MetS and obesity-related indices with △eGFR using multivariable linear regression analysis in follow-up participants (*n* = 27,033).

Obesity-Related Indices	Multivariable	
	Unstandardized Coefficient β (95% Confidence Interval)	*p*
MetS ^a^	−1.972 (−2.569, −1.374)	<0.001
BMI (per 1 kg/m^2^) ^b^	−0.064 (−0.135, 0.007)	0.075
WHR (per 1%) ^b^	−0.030 (−0.068, 0.008)	0.126
WHtR (per 1%) ^b^	−0.058 (−0.100, −0.016)	0.007
LAP (per 1) ^b^	0.022 (0.011, 0.033)	<0.001
BRI (per 1) ^b^	−0.206 (−0.339, −0.072)	0.002
CI (per 0.1) ^b^	−0.452 (−0.740, −0.163)	0.002
BAI (per 1) ^b^	−0.060 (−0.123, 0.004)	0.066
AVI (per 1) ^b^	−0.129 (−0.206, −0.051)	0.001
VAI (per 1) ^c^	−0.174 (−0.318, −0.031)	0.017
ABSI (per 0.01) ^b^	−0.585 (−1.046, −0.125)	0.013

Values expressed as unstandardized coefficient β and 95% confidence interval. Abbreviations are the same as in Table 1. ^a^ Adjusted for age, sex, hemoglobin, total cholesterol, LDL cholesterol, and uric acid (significant variables of Table 5 except for diabetes, hypertension, systolic and diastolic blood pressures, fasting glucose, and HDL cholesterol). ^b^ Adjusted for age, sex, diabetes, hypertension, systolic and diastolic blood pressures, fasting glucose, hemoglobin, total cholesterol, HDL cholesterol, LDL cholesterol, and uric acid (significant variables of Table 5). ^c^ Adjusted for age, sex, diabetes, hypertension, systolic and diastolic blood pressures, fasting glucose, hemoglobin, total cholesterol, HDL cholesterol, LDL cholesterol, and uric acid (significant variables of Table 5 except for HDL cholesterol).

## Data Availability

The data underlying this study are from the Taiwan Biobank. Due to restrictions placed on the data by the Personal Information Protection Act of Taiwan, the minimal data set cannot be made publicly available. Data may be available upon request to interested researchers. Please send data requests to: Szu-Chia Chen, PhD, MD. Division of Nephrology, Department of Internal Medicine, Kaohsiung Medical University Hospital, Kaohsiung Medical University.

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
