# Peer review of "Metabolic Syndrome and Obesity-Related Indices Are Associated with Rapid Renal Function Decline in a Large Taiwanese Population Follow-Up Study"

_biomedicines, 2022, doi:10.3390/biomedicines10071744_

Round 1

Reviewer 1 Report

No further comments.

Author Response

No further comments.

Ans: Thank you for your reviewer to make our manuscript better.

Reviewer 2 Report

The mauscript has been sufficiently revised in some core passages. The authors to rephrase the introduction to furnish a wider introductory context and better substantiate the need for this study. For example what is the link between obesity/dysmetabolism and the decline of renal function? 

Author Response

The manuscript has been sufficiently revised in some core passages. The authors to rephrase the introduction to furnish a wider introductory context and better substantiate the need for this study. For example, what is the link between obesity/dysmetabolism and the decline of renal function?

Ans: Thank you for your comments. We have added the link between obesity/dysmetabolism and the decline of renal function in introduction.

  • Obesity is closely related to metabolic alteration in the body. According to previous study, the mechanisms involved are related to the interaction between adipocytes and immune cells [8]. The increase in adipocyte size and the inability to store triglycerides would cause the development of an inflammatory response, which is thought to be associated with the formation of metabolic disorders. This would promote inflammation and activate apoptotic pathways. Inflammation of adipocytes affects insulin signaling and metabolism in adipocytes through autocrine effects of inflammatory cells, and endocrine effects of adipokines affect insulin sensitivity in other tissues, resulting in local or even systemic insulin resistance [9]. Insulin resistance may further cause renal injury through renin-angiotensin system activation, increases in aldosterone and angiotensin II with subsequent effects on endothelin-1 and insulin-like growth factor-1, and the generation of reactive oxygen species [10,11]. (Line 83-95)

Round 2

Reviewer 2 Report

the introduction still needs to bee improved. It's unclear how obesity and metabolic syndrome generate CKD.

Author Response

the introduction still needs to be improved. It's unclear how obesity and metabolic syndrome generate CKD.

Ans: Thank you for your comments. We have added the issue about how obesity and metabolic syndrome and kidney disease in Introduction to make readers more clear.

  • The exact mechanism underlying the association between MetS and obesity with kidney disease has yet to be fully elucidated, although insulin resistance has been proposed. Insulin resistance can cause renal injury through renin-angiotensin system activation, increases in aldosterone and angiotensin II with subsequent effects on endothelin-1 and insulin-like growth factor-1, and the generation of reactive oxygen species [11,12]. In addition, tissue resistance to insulin has been shown to lead to a reduction in nitric oxide production, which was associated with impaired tubulo-glomerular feedback, hyperfiltration, and sodium retention, leading to disruption of the autoregulation of renal blood flow and glomerular filtration [13]. Obesity can also affect the production of adipokines, leading to renal dysfunction. Altered levels of adipokines, such as leptin, adiponectin, resistin, and visfatin have been shown to decrease the GFR and increase albuminuria by increasing glomerular permeability, fusing podocytes, mesangial cell hypertrophy, and interfering with tubular networks [14]. (Line 99-112)
